# Comparing World City Networks by Language: A Complex-Network Approach

**Wenjia Zhang** [1,2,*] **, Jiancheng Zhu** [2] **and Pu Zhao** [2]

1   Urban Future Laboratory of Peking University (Shenzhen), Peking University Shenzhen Graduate School, Shenzhen 518055, China
2   School of Urban Planning & Design, Peking University Shenzhen Graduate School, Shenzhen 518055, China; zhujc@pku.edu.cn (J.Z.); zhao-pu@pku.edu.cn (P.Z.)
*   Correspondence: zhangwj@pkusz.edu.cn; Tel.: +86-755- 26032134

**Abstract:** City networks are multiplex and diverse rather than being regarded as part of a single universal model that is valid worldwide. This study contributes to the debate on multiple globalizations by distinguishing multiscale structures of world city networks (WCNs) reflected in the Internet webpage content in English, German, and French. Using big data sets from web crawling, we adopted a complex-network approach with both macroscale and mesoscale analyses to compare global and grouping properties in varying WCNs, by using novel methods such as the weighted stochastic block model (WSBM). The results suggest that at the macro scale, the rankings of city centralities vary across languages due to the uneven geographic distribution of languages and the variant levels of globalization of cities perceived in different languages. At the meso scale, the WSBMs infer different grouping patterns in the WCNs by language, and the specific roles of many world cities vary with language. The probability-based comparative analyses reveal that the English WCN looks more globalized, while the French and German worlds appear more territorial. Using the mesoscale structure detected in the English WCN to comprehend the city networks in other languages may be biased. These findings demonstrate the importance of scrutinizing multiplex WCNs in different cultures and languages as well as discussing mesoscale structures in comparative WCN studies.

**Keywords:** world city network (WCN); language; complex network; mesoscale structure; webpage big data; weighted stochastic block model (WSBM)

## 1. Introduction

World cities have been treated as strategic sites in the global economy and society, interlinked by multiplex relations for accumulating wealth, control, and power through the flows of capital, information, people, and knowledge [1–4]. World city network (WCN) studies have become an important approach for understanding globalization, which involves not only an economic process of global capital accumulation and articulation, but also cultural, political, and social processes [5]. Facing such 'multiple globalizations', the focus of WCN studies has shifted from a singular linkage between cities to multiple types of intercity relations [6].

However, only a limited number of studies have investigated multiple globalizations and compared multiplex WCNs. Among them, a thread of research focuses on variant types of economic linkages between world cities by looking at different corporate connections [5,7,8]. They survey multiple city-by-firm networks by comparing the multinational companies of advanced product services (APS) or other industries. Another thread of studies highlights the dynamics of corporate WCNs driven by economic globalization in different periods [4,6]. However, compared to economic WCNs, non-economic WCNs are greatly under-researched [9,10]. In particular, few studies investigate and compare the differences of WCNs in the views of varying cultures and languages.

To fill this gap, this paper aims to expand our understandings of multiple globalizations by comparing WCNs reflected in the Internet webpage content in different languages. In such WCNs, the strength of the relation of two cities is defined as the number of webpages with both cities' names. The weight of intercity linkages is the co-occurrence frequency of cities collected by the Advanced Google Search Engine from May to June 2016. Based on the webpages of different languages, which here includes English, French, and German, we can derive three WCNs by language.

The comparison of WCNs relying on webpage languages can better reveal variant city network patterns in different cultures. First, language is an important carrier of culture; the distribution and visibility of languages are closely related to ethnicity, culture, socioeconomic status, and political power [11,12]. Second, the webpage data as a virtual mapping of the real world have been increasingly used for the analysis of geographical imagination [13–16]. If two cities are frequently co-occurred on webpages, it probably implies that they are closely connected, and their connection is comprehensive. These two cities may have strong economic connections as well as non-economic ties, such as through the flow of tourists and migration and the similarities in history, socio-politics, and culture [15,17]. Therefore, the WCNs "textualized" by different languages could be quite different; the imagination of globalization varies with our culture and the language we use offline and online. Additionally, while cultural linkages among cities are often difficult to be quantified, language-based textual data provide a quantitative basis for comparing the global geographic imaginations from different cultural groups [13,14].

In addition, comparative studies of multiple WCNs need methodological innovation. Most comparative studies emphasize the macroscale features of WCNs, such as city rankings and distributions of centralities, along with the visualization techniques on network and geographic layouts [5,6,14,18]. The macroscale analysis cannot unravel the inner structure of a WCN and fails to answer some critical questions [5,6,14,18], for example, whether the WCN has a core–periphery or regionalization structure, or whether a city belongs to a core or a periphery? To better investigate these questions, we need to detect the mesoscale structures in WCNs, that is, the city clustering properties based on the patterns of intercity linkages. As conceptualized in Zhang and Thill [14], many potential mesoscale structures could exist in WCNs, including a community structure showing territorialism or regionalization, a core–periphery structure representing city hierarchies, a random or a flat-world structure, and hybrid structures.

There are many approaches to city clustering, including principal components and hierarchical clustering analysis based on city nodal attributes [4,19] and network-based community detection and block models [20–23]. However, these approaches could be easily trapped into the "methodological determinism" [14], in which the method used for clustering may pre-determine a grouping structure and pre-exclude the "true" mesoscale structure. For example, community detection algorithms can find out "communities" in a WCN, but difficult to detect "cores" and "peripheries". If a WCN truly has a core–periphery structure, the application of community detection methods for city clustering may detect a biased mesoscale structure. Rozenblat et al. is among the first to compare the community structures or the multipolar regionalization of cities in four types of multinational firm networks [21], including the corporate networks of high technologies, low technologies, knowledge intensive services, and less knowledge-intensive services. Martinus and Sigler compared the community (clustering) structures in the city networks formed by the locational links of varying types of firms listed on the Australian Securities Exchange [20], including energy, material, industrial, and financial corporations. However, both studies above presume the mesoscale structures are community structures in advance.

In order to solve these analytical issues, this study applied a novel complex network approach—that is, a weighted stochastic block model (WSBM) [14,24]—to infer the latent mesoscale structures of WCNs in different languages and created a probability-based index for quantitatively comparing the (dis)similarity between WCNs. This approach is also innovative for comparison. While existing comparative methods can establish whether

two WCNs are different [7,8,25], they rarely provide a quantitative measure to identify how large the difference between the two WCNs is.

The contributions of this paper are thus twofold. First, in order to eliminate the analytical biases from existing city clustering methods, such as community detection and principal component analysis, we developed a WSBM method to disentangle the latent mesoscale structures (e.g., core–periphery versus communities) in variant WCNs and provided a quantitative measure for mesoscale comparison. Second, this study is among the first to distinguish WCNs across scales reflected in different languages, helping us quantify the cultural differences in understanding global urban hierarchies and networks under the context of multiple globalizations.

## 2. Related Work

An increasing amount of empirical literature has investigated the comparison of multiple WCNs. For example, most of the research focuses on variant types of economic linkages between world cities by looking at different corporate connections [5,7,8], including the connections of advanced producer services (APS) companies and other multinational firms, as well as the mobility links by air, shipping, and land transportation. For example, Krätke compared the similarities and differences of WCNs of multinational manufacturing industries, such as pharmaceutical and biotechnology companies [7], automotive industry companies, and technical hardware and equipment companies [25]. By comparing the network features, these studies found significant differences in leading cities and city rankings in different industries. Martinus and Tonts reported that the WCN connected by energy industries shows a new pattern of subgroup characteristics; the world energy network consists of three "meta-networks" in Europe, America, and the Asia-Pacific region [26]. Sigler and Martinus further distinguished urban networks defined by varying industrial sectors, including material, energy, industrial, and financial sectors [8].

Some comparative WCNs research focuses on the dynamic evolution of corporate WCNs in different periods. For example, some scholars analyzed the changes in network connectivity and ranking of cities by comparing multiple-year APS enterprise datasets collected by the group of Globalization and World Cities Research Network (GaWC) [6]. They found that the overall levels of network connectivity have grown by years, and the connectivities of a few city clusters have increased significantly. Derudder and Taylor recently delineated three stages of globalization to date (i.e., extensive, intensive and Chinese globalizations) by investigating the changing global distributions of APS companies [4]. Among the three globalizations, the extensive globalization presents a core–periphery structure of the hierarchical dependence between developed and developing countries [27] (pp.1651–1663), fitting into the "world city hypothesis" [1]. This intensive globalization rather represents a process of mutual agglomeration between demand (corporate headquarters) and supply (APS) anchored in "global cities" [28] (pp.170–172), while the Chinese globalization shows that China has enhanced its influence on globalization.

While most previous studies emphasize on economic connections among world cities, they have aroused criticisms of overlooking non-economic linkages [9,10]. Therborn claimed that the existing research relates most to economic globalization but neglects to understand globalized cities from the perspective of culture and history [10]. Some existing studies have attempted to survey such an economic bias by looking at various non-economic intercity relations, in terms of non-governmental organizations (NGO) [18], higher education institutions [29], media companies [30], and textual contents or global news by topics [13,17].

Only a limited number of studies have investigated the WCN variations reflected in varying languages and cultures. Some existing research has demonstrated that the intercity linkages in WCNs are associated with the cultural environment of those connected cities. For example, the analysis of the Islamic financial services (IFS) sector in the world found that the characteristics of the WCN under the Islamic culture significantly differ from the geographic imagination in the Western countries [31]. Another study based on the data of

corporations listed on the Australian Securities Exchange found that cultural proximity (such as history and language) has a significant impact on shaping the relationship between world cities [20].

## 3. Data from Web Crawling

For measuring city connectivity reflected in different languages, this study used a webometrics approach to collect intercity linkage data [32]. We used the Advanced Google Search Engine in the Google search platform to collect the co-occurrence frequency of two cities by entering two cities' names in different languages. The underlying assumption is that if two cities appear on the same webpage, it implies that they probably have a conceptual linkage with shared function and value under a general cognitive linguistic framework, and the aggregate number of co-occurrences is the weight of their connectivity. The appearance of two cities on web pages may reflect multiple types of linkages between them, including economic, transportation, tourism, historical, social, cultural, and political relationships [15,17]. This study selected English, German, and French from the world's 10 most influential languages [33]. Additionally, the website estimates the ranking of content languages from the top 10 million websites. As of early 2016, the top content language for websites is English, accounting for 55.5% of all webpages, while the corresponding proportions of German and French are 5.8% and 4%, respectively, ranking no. 3 and no. 6 among about 100 languages in the world.

Before web crawling, several steps are needed to identify world cities and their names for searching. First, since the selection of cities may bias WCN studies [9], we identify 126 world cities in terms of a survey of existing literature. If cities occur at least twice among 22 influential WCN studies in recent 30 years, as articulated in Son [34] (pp.145–178) and Zhang and Thill [14], they are selected in this study. The 126 world cities consist of 42 cities in Asia (33.3%), 41 in Europe (32.5%), 22 in North America (17.5%), 11 in South America (8.7%), 7 in Africa (5.6%), and 3 in Oceania (2.4%). Second, Google Search needs one to enter keywords of the toponym, which is thus likely to cause ambiguation issues, such as the duplication or ambiguity of city names [14]. For reducing ambiguity, we combined city names and corresponding country names as keyworks for the search. Third, after identifying English names of a city and country, we translated all of them into German and French by native speakers in these two countries and used the Advanced Google Search platform that allows for a search by language.

Lastly, a Python-based web-crawling algorithm was developed to automatically search and crawl the number of co-occurrences of any two cities. For reducing the bias caused by the inherent instability of Google Search, we collected 28 sets of co-occurrence data from 9 May to 6 June 2016 and conducted 441,000 searches (28*126*126) per language. Then, we chose the maximum co-occurrence number in the 28 sets of data to approximate the intercity relational ties. Finally, a total of 0.138 trillion webpages were searched, including 0.120 trillion English webpages (87.3%), 12.57 billion German webpages (9.1%), and 4.92 billion French webpages (3.6%).

## 4. Methods: The Complex-Network Approach

This study adopted a comprehensive approach to comparing microscale (i.e., local), macroscale (i.e., global), and mesoscale (i.e., intermediate) features of WCNs in varying languages. The field of network science has well defined the comprehensive network properties into three scales, including micro, macro, and meso scales [35–37]. In particular, microscale features in WCNs measure the centrality and power of individual cities as well as the influence of city-dyads, while the macroscale structure captures the overall features of each WCN by investigating the ranking and distribution of local attributes. For example, degree centrality is a typical node-based microscale feature for measuring nodal importance, indicating the sum of weights of linkages connected with a city node. As most existing WCN studies only look at the microscale and macroscale features in WCNs but overlook the mesoscale [25,29,38,39], this study particularly focused on the

comparison of mesoscale structures in WCNs, which describe the grouping features of city nodes based on intercity relational ties by using the WSBM developed by Aicher et al. [35] and a comparative mesoscale analysis developed by Zhang and Thill [14].

In fact, many methods have been developed to detect mesoscale structures in networks. Early analysis is oriented from Social Network Analysis (SNA) with a focus on finding cohesive subgroups, within which social actors are more closely, mutually, or densely connected than those in the rest of the network [40] (pp.345–391). Corresponding techniques, such as k-cores, k-cliques, k-components, k-clan, k-club, and k-plex [40] (pp. 345–391), [41], search for maximal sub-networks with k as a threshold to screen nodes or edges based on their properties. This genre of analysis only partially reveals the mesoscale structure because it identifies some locally dense subgraphs but neglects other nodes and edges outside the cohesive subgroups [41,42].

Recent studies have turned to the investigation of specific mesoscale structures. Among them, a considerable amount of literature has contributed to the algorithmic detection of communities [42–44]. A popular approach to community detection relies on modularity optimization [45], a quality function for evaluating whether a partition of a network is good or bad by comparing the partition with a null model without communities (e.g., a random network). This approach is often superior to some traditional methods, such as graph partitioning, partitional clustering, and hierarchical clustering, since it provides not only an effective partition but also assesses the goodness of the partition. While an exhaustive optimization of modularity is probably impossible [46], many heuristic algorithms, such as greedy algorithms, simulated annealing, and evolutionary algorithms, have been advanced to find good approximations of the modularity maximum [47–49]. However, most community-detection methods postulate the existence of communities in a network and exclude other possible and hybrid mesoscale structures.

By contrast, a branch of literature focuses on the core–periphery (CP) structure [41,50,51]. As first concretely defined by Borgatti and Everett [50], a CP structure in a binary network has 1-blocks (with all matrix cells as 1) in the core, 0-blocks (with all 0) in the periphery, and blocks mixed with 1 and 0 in the core/periphery position. They further extended the CP measure from binary to weighted networks by computing a "coreness" value for each node (i.e., the level of core versus peripheral position) using the Minimum Residual (MINRES) algorithm. Rombach et al. generalize Borgatti and Everett's measures to compute a structure of multiple cores and further evaluate the quality of the cores [51]. Kostoska et al. developed a LARDEG (largest degeneracy) model to detect the core–periphery structure in sectoral international trade networks across countries [52]. However, these CP-detection methods have similar limitations of community detection algorithms since they rule out other potential mesoscale structures.

In order to avoid the predetermined grouping structure (i.e., the trap of "methodological determination" [14]), we adopted the WSBM approach without predefining any block structures. Intuitively, WSBM aims to maximize the likelihood function by inferring the grouping $z$ and stochastic block matrix $\theta = [\theta_{kk'}]_{K \times K}$, where $K$ denotes the number of groups of cities. Given an adjacency matrix $A = \{a_{ij}\}$ of the city network ($a_{ij}$ is the dyadic weight from cities i to j) and presuming $\theta$ is normally distributed, with the mean $\mu = [\mu_{kk'}]_{K \times K}$ and variance $\sigma^2 = [\sigma^2_{kk'}]_{K \times K}$, the likelihood function is derived as follows [14]:

$$P(A|z,\theta) = P\left(A|z,\mu,\sigma^2\right) = \prod_{i,j} \exp\left( a_{ij} \frac{\mu_{z_i,z_j}}{\sigma^2_{z_i,z_j}} - a_{ij}^2 \frac{1}{2\sigma^2_{z_i,z_j}} - \frac{\mu^2_{z_i,z_j}}{2\sigma^2_{z_i,z_j}} - \log \sigma_{z_i,z_j} \right) \quad (1)$$

This study adopted a Bayesian regularization approach developed by Aicher et al. [35]. One can treat parameters as random variables by assigning an appropriate prior distribution $P(z,\theta)$. Then, we can obtain the posterior distribution $P(z,\theta|A)$ by Bayes' law:

$$P(z,\theta|A) \propto P(A|z,\theta)P(z,\theta) \quad (2)$$

Since it is difficult to analytically calculate the posterior distribution, a factorizable distribution $q(z, \theta) = q_z\ (z)q_\theta\ (\theta)$ is developed to approximate $P(z, \theta|A)$ with a machine learning method. The WSBM can easily determine the optimal number of groups without an arbitrary guess. It does not define the structure in advance as the conventional block model does, which help find some new mesoscale structures that are not found in the past but may truly exist in WCNs. This study used Matlab 2016a (The MathWorks, Natick, MA USA) to run the WSBM algorithm and provides visualizations in both Matlab and an open-source software Gephi 0.9.1 (www.gephi.org, accessed on 12 December 2020). Here, the input of the WSBM is an adjacency matrix of a 126-by-126 city network. The output is the partition results (i.e., a cluster of cities into groups), as shown in the heatmap and block matrix.

Although Zhang and Thill applied the WSBM to detect mesoscale structures reflected in the English webpages [14], they did not compare WCNs in different languages. This paper extended to focus on the comparison of mesoscale structures in English, German, and French. We defined an Incremental Fit Index (*IFI*) for the comparison of geographic imagination in different languages at the meso scale. In particular, $IFI_{xy}$ represents to what extent a given partition $x$ for comparison approximates the optimal partition y detected by the WSBM. The *IFI* value ranges from 0 to 1. If $IFI_{xy} = 0$, the partition $x$ is close to a null model (without partition, K = 1). If $IFI_{xy} = 1$, the partition $x$ is equal to the optimal partition $y$. In other words, the network is more likely to have a mesoscale structure of the partition $x$ if the *IFI* value is larger. The equation of $IFI_{xy}$ can be written by:

$$IFI_{xy} = \frac{LL_x - LL_{null}}{LL_y - LL_{null}} \tag{3}$$

Here, $LL_{null}$ is the log-likelihood score without any partition structure (i.e., K = 1), as computed in Equation (1); $LL_y$ is the log-likelihood value of the optimal partition $y$ estimated; $LL_x$ is the log-likelihood value of the given partition $x$ for comparison.

## 5. Results

### 5.1. Comparative Macroscale Analysis: Rankings and Distributions of Degree Centralities

This study first provided a comparative macroscale analysis of city ranking and distributions of microscale measures. Figure 1 shows the geographic distribution of 126 world cities, their degree centralities and top city dyads, as reflected in the webpages in English (Figure 1a), German (Figure 1b), and French (Figure 1c). In each figure, the sizes of nodal markers are normalized levels of degree centralities in each network, with the top city (i.e., New York) setting at 1, and the colors of the lines show city dyads in different rankings. The geographic distribution of WCNs has some similarities as well as variant features in the three languages. First, for example, the central cities with top centralities of the WCNs in the three languages are mostly located in North America and Europe. However, the centralities (i.e., the size of nodes in Figure 1) are more evenly distributed across the world in English and French, while those in the German WCN are more concentrated in the German cities (e.g., Berlin and Frankfurt) and top world cities such as New York, London, and Paris. Compared to the WCN in English, cities in the French and German WCNs are more intensively distributed in the European region than the American region.

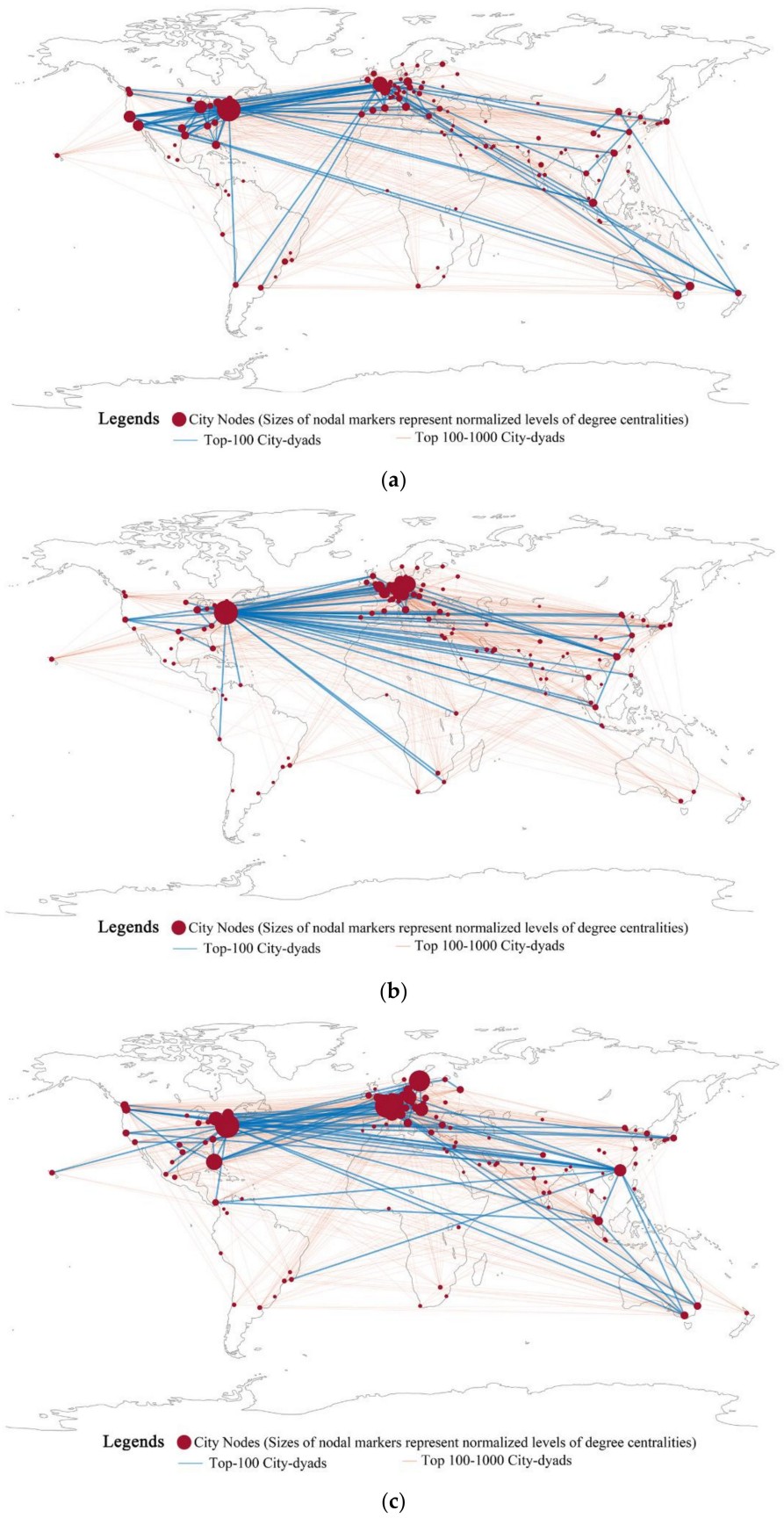

**Figure 1.** Geographic distributions of world city nodes and dyads in three languages: (**a**) English, (**b**) German, (**c**) French.

Second, the distributions of the top 100 and top 500 city dyads seemingly show that the English and French WCNs are more globalized because the top dyads connect most continents and important cities around the world, while the top city dyads of the German network are sparsely connected to South America and Oceania.

Third, supposing that the networks of top city dyads represent the primary geographic imaginations of globalization, all the three population groups using English, German, and French have similar globalization visions of a triadic configuration connecting the US, Europe, and coastal area of East Asia. In particular, the aggregate connections between Europe and North America are the strongest in all the three networks, while the linkages connecting Asia with North America and Europe also rank in the top. In edge areas such as South America, different languages have their distribution centers, such as Buenos Aires and Santiago in the English network, Caracas and Lima in the German network, and Rio de Janeiro and Panama City in the French network.

Table 1 reports the rankings of city nodes with the top degree centralities. In the English WCN, New York, London, and Chicago rank in the top three. The top 20 world cities are mostly the ones using English as the primary official language, and most of them locate at the Commonwealth and the US. The exceptions are Paris, Berlin, Singapore, Rome, and Barcelona, which are traditional important world cities using other languages. Similarly, in the German WCN, many top 20 cities are German-dominant cities, and the top 10 cities are mainly German cities except New York, London, and Paris. In the French WCN, although the top 20 cities are mostly French-speaking cities, they are not concentrated in France but are widely distributed in North America and Europe.

**Table 1.** World city rankings of the top 20 degree centralities in three languages.

| Rank | English | | German | | French | |
|------|---------|--------|--------|--------|--------|--------|
| | **World City** | **Degree** | **World City** | **Degree** | **World City** | **Degree** |
| 1 | New York | 17.80 | New York | 1.96 | New York | 0.55 |
| 2 | London | 10.33 | Berlin | 1.27 | Paris | 0.52 |
| 3 | Chicago | 8.46 | Frankfurt | 0.98 | Stockholm | 0.47 |
| 4 | San Francisco | 7.62 | Hamburg | 0.83 | Miami | 0.34 |
| 5 | Los Angeles | 6.84 | Cologne | 0.81 | Cologne | 0.31 |
| 6 | Paris | 6.44 | London | 0.74 | Toronto | 0.29 |
| 7 | Boston | 5.43 | Munich | 0.74 | Lyon | 0.29 |
| 8 | Berlin | 4.88 | Paris | 0.73 | Berlin | 0.27 |
| 9 | Melbourne | 4.85 | Stuttgart | 0.68 | Budapest | 0.26 |
| 10 | Sydney | 4.83 | Dusseldorf | 0.52 | Hong Kong | 0.24 |
| 11 | Toronto | 4.69 | Bonn | 0.48 | London | 0.22 |
| 12 | Singapore | 4.58 | Chicago | 0.41 | Montreal | 0.21 |
| 13 | Miami | 4.58 | Rome | 0.41 | Amsterdam | 0.18 |
| 14 | Dallas | 4.51 | Hong Kong | 0.41 | Milan | 0.16 |
| 15 | Philadelphia | 4.47 | Birmingham | 0.31 | Copenhagen | 0.16 |
| 16 | Houston | 4.31 | Edinburgh | 0.30 | Brussels | 0.15 |
| 17 | Rome | 4.29 | Singapore | 0.29 | Singapore | 0.15 |
| 18 | Barcelona | 3.81 | Budapest | 0.29 | Seattle | 0.14 |
| 19 | Washington DC | 3.65 | San Francisco | 0.29 | Rome | 0.14 |
| 20 | Atlanta | 3.62 | Istanbul | 0.27 | Birmingham | 0.14 |

The unit of degree centrality is a billion webpages.

By looking at the microscale features of specific cities, we found that New York ranks first in the WCNs of all the three languages, while the second place is the capitals of the UK, Germany, and France, respectively. This finding suggests that the dominant global influence of New York crosses cultural and language boundaries. The cities of Paris, London, Berlin, Rome, and Singapore appear three times on the ranking lists, showing their global influences in different cultures. Additionally, Hong Kong ranks top 20 in the French and German WCNs but not in the top 20 rankings of the English WCN. This implies that Hong Kong appears to play a more important role in the French and German

worlds than the English world. It is also interesting that no Chinese or Japanese cities appear on the three lists in Table 1, although many studies reveal that Chinese cities play an increasingly important role in global economic development and in shaping the corresponding WCNs [4,53]. It is reasonable that Chinese cities may have very large economic linkages with the Western cities but relatively weak connections in culture or other noneconomic properties. These findings demonstrate that the influence of a world city may vary across language and culture, by which people may perceive different world geography and globalization. It is thus important to scrutinize the roles and positions of world cities in multiplex networks.

Figure 2 presents the distributions of aggregate dyadic weights across city nodes and city dyads of WCNs in three languages (Figure 2a,b), along with the plots of cumulative degree distribution (Figure 2c). The cumulative distribution of degree or weight $C(k)$ represents the fraction of nodes with degree smaller than $k$ or the fraction of edges (i.e., city dyads) with edge weight smaller than $k$, they are often used to analyze whether degree centralities or edge weights are concentrated in a few top nodes/edges or whether they fit a power-law distribution. According to Figure 2a, in general, the weight distributions across city nodes look similar in the three networks. However, for top-ranking cities, the German and French WCNs appear more hierarchical than the English WCNs: the top 10% of world cities accumulate about 30% of the total edge weights in the German and French WCNs, but about 20% of the weights in the English WCNs. On the contrary, the top 40% of cities occupy about 80%, 70%, 60% of total weights in the English, French, and German WCNs, respectively. Additionally, the weight distributions across dyads (Figure 2b) are more polarized and vary less with language, with about 80% of edge weights distributed at the top 10% city dyads. Figure 2c further shows the cumulative degree distribution. Degree centralities do not simply follow a power-law distribution because their distributions are not linear in all three languages. For the English and German WCNs, the degree distribution curves can be separated into two straight lines that fit the power law, respectively.

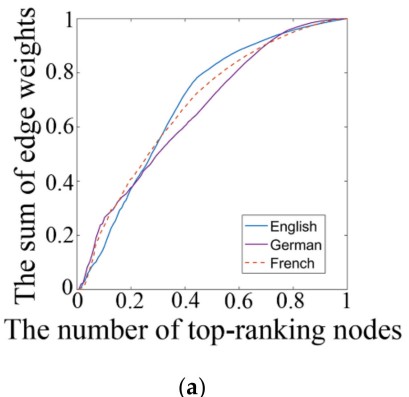
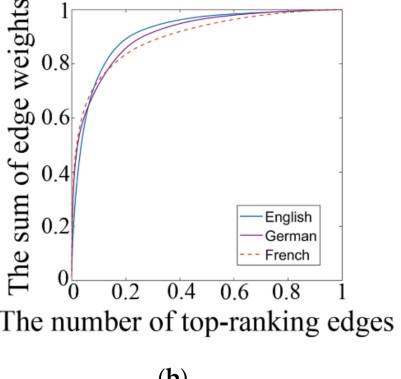
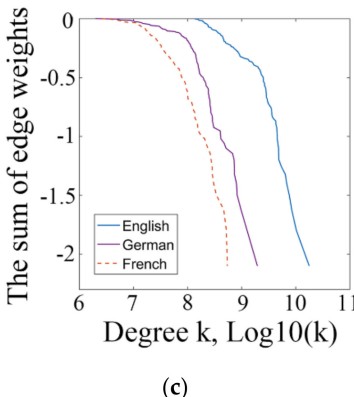

(a)            (b)            (c)

**Figure 2.** Distribution of dyadic weights and nodal degree centralities (**a**) across city nodes and (**b**) across city dyads, and (**c**) cumulative distribution of degree centralities.

*5.2. Comparing Mesoscale Structures Based on WSBM Findings*

Although the macroscale analysis of local measures above provides rough information about the mesoscale structure of a WCN, it implies nothing about the number of clusters/groups of cities in the WCN and their roles in networks. This study applied the WSBM to provide a statistical inference of the number of groups and the optimal mesoscale structure. We first estimated a set of WSBMs by changing the number of partitions K from 2 to 20, then checked how the marginal log-likelihood values change with K and detected the optimal number of partitions when the corresponding log-likelihood value is maximized. After these processes, the WSBM results show that the optimal number of groups in the English, German, and French WCNs are 8, 6, and 6, respectively.

Since the WSBM does not presume a mesoscale structure in the WCNs, we might detect variant grouping results from the heatmaps (Figure 3) and block matrices (Figure 4). Figure 3 shows the heatmaps of the 126 × 126 adjacency matrix partitioned by the WSBM results in different languages (left column) and visualizes the WCNs in a Fruchterman–Reingold layout by Gephi (right column). Figure 4 reflects the intra- and inter-group relations, which help determine the role of each group as a core, semi-core, semi-periphery, or a periphery. Here, a core represents a group of cities with strong connections within the group and with cities in other groups, while a periphery is a group of cities weakly connected with each other and the other groups. The grouping and roles of world cities in different languages is provided in detail in the Appendix A (Table A1,Table A2,Table A3).

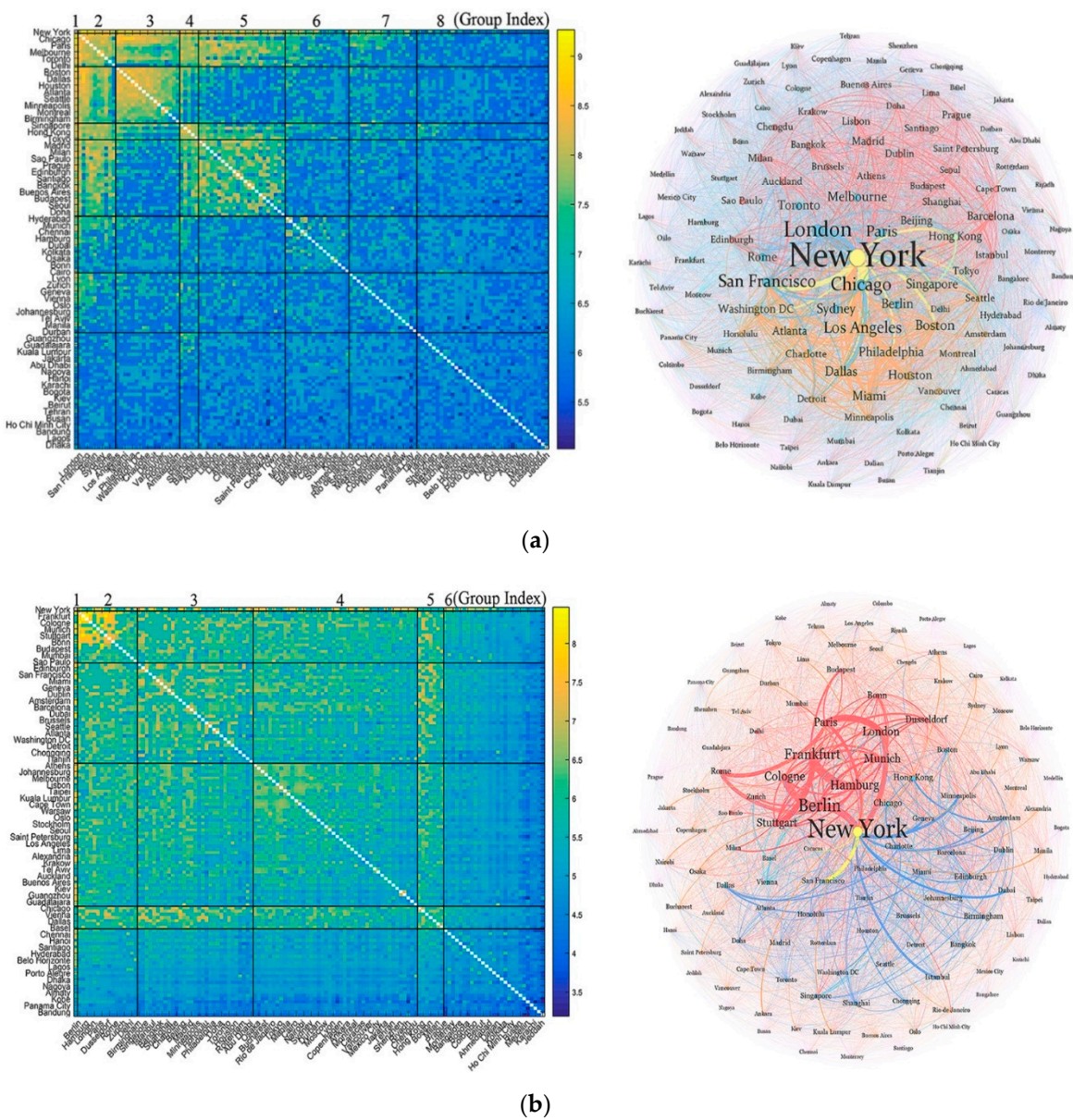

(**a**)

(**b**)

**Figure 3.** *Cont.*

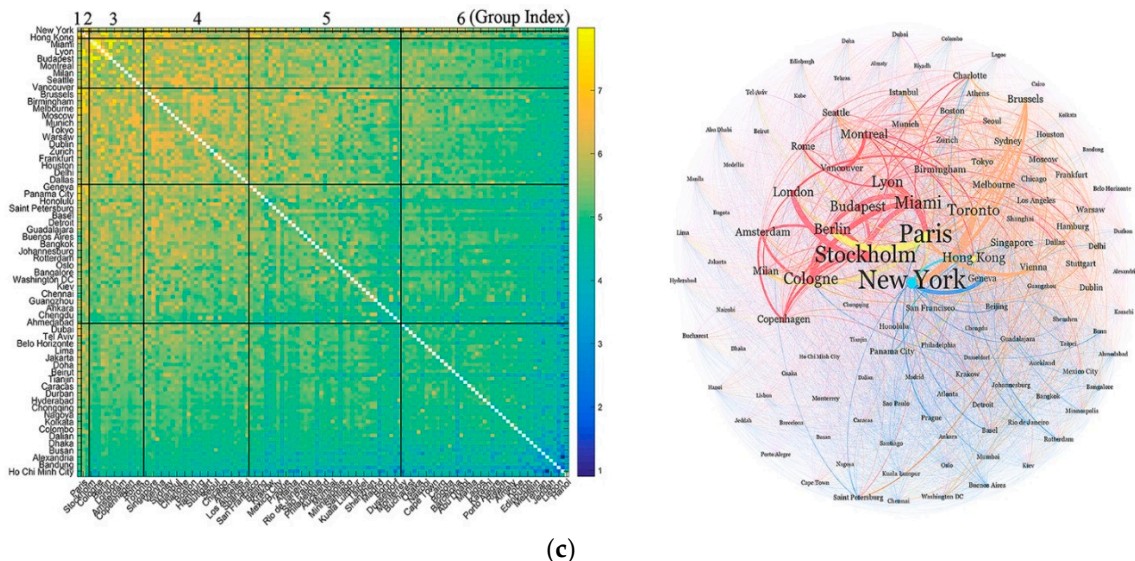

(**c**)

**Figure 3.** Heatmaps and networked layouts of WCNs in different languages: (**a**) multicores-peripheries structure (English), (**b**) dual-cores-peripheries structure (German), (**c**) a typical core-periphery structure (French).

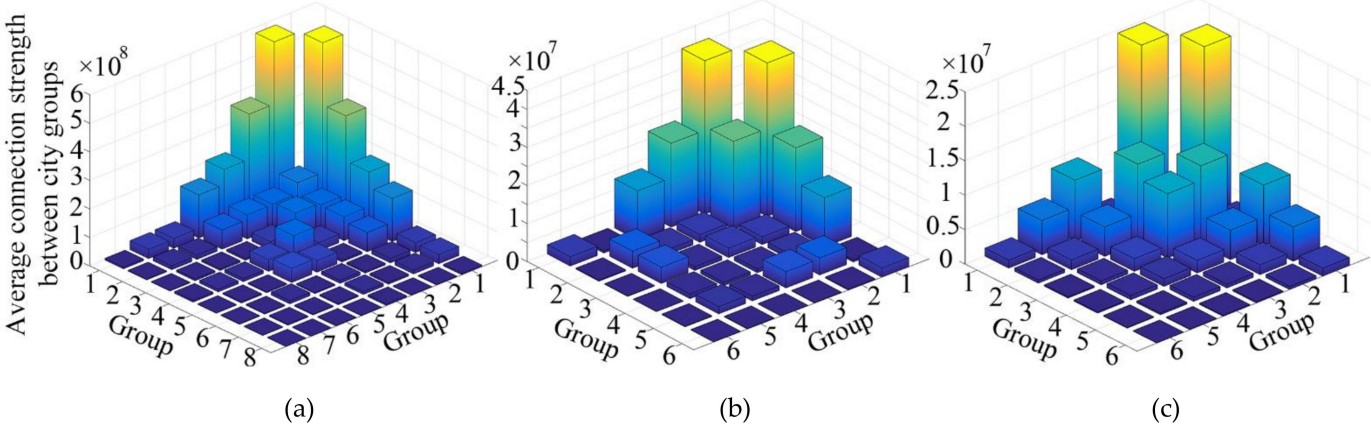

(a)  (b)  (c)

**Figure 4.** The block matrix of mesoscale structures: (**a**) English, (**b**) German, (**c**) French.

According to Figure 3, the mesoscale structures are saliently variant in different languages. For example, the English world perceives a hybrid network of community and core–periphery structures (Figure 3a) or a multicores–periphery structure [14]. In more detail, the English WCN has three cores: a global core with global influence (i.e., Group 1 with only one global city, that is, New York) and two regional cores (Groups 2 and 4) with macro-regional influences in the Western and Asian areas, respectively. Groups 3 and 5 play a semi-periphery role serving the first two cores and the Asian cores (i.e., Group 3). Groups 6, 7, and 8 are peripheral since they have strong connections with the three core groups but relatively weak linkages with other periphery groups. The mesoscale structure is better recognized in Figure 4a, which provides the 3-dimensional heatmap of block matrix by partition.

The German WCN has two specific cores, but is without community structures (Figures 3b and 4b). Group 1 (i.e., New York) is the global core while Group 2 (including many cities in Germany and surrounding countries) is the regional core. The rest groups are peripheral. In particular, Groups 3, 4, and 6 have strong connections with the global core but relatively weak links with the second core. By contrast, only Group 5 serves as the peripheral role of the regional core (i.e., Group 2). Unlike the English WCN, no peripheries serve for both cores in the German WCN.

The French WCN rather presents a mesoscale structure more like a typical core–peripheral structure (Figures 3c and 4c). The first three groups are the three cores at different levels while the rest three groups are peripheral, with relatively much stronger connections with the three cores than the peripheral groups.

By comparing the mesoscale structures of WCNs in three languages, we can find that New York is the global center no matter in which languages, and Paris shows up twice as the macroregional core city (in English and French). However, the roles of many world cities vary across language and culture. For example, Hong Kong plays a very influential role as a global center in the French world and as a regional center in the English network, while it is relatively at the edge in the German network. In addition, cities are prone to become a cluster when they are culturally proximate, such as when the first languages of these cities are the same, as shown in Appendix A.

Moreover, we compared the similarities of mesoscale structures in different languages by using the *IFI* value calculated in Equation (3). This comparative approach can quantify the overall difference or similarity between a given mesoscale structure (i.e., a city partition) and the optimal partition estimated by the WSBM. Table 2 tabulates the results of *IFI* values for comparison. We first examined to what extent the partition of continent-based regionalization (i.e., six groups by continent) approximates the detected mesoscale structures of WCNs in three languages. Based on the *IFI* values, the German and French WCNs are more likely to exhibit a mesoscale structure of continent-based regionalization than the English WCN, although their corresponding *IFI* values are all below 50%. This finding suggests that the WCN reflected in the English webpage contents has less significant territorialist properties, while territoriality may play a more important role in shaping people's geographic imagination of globalization in the French and German WCNs.

**Table 2.** $IFI_{xy}$ Values.

| Partition $x$ \ Optimal Partition $y$ in | Continent-Based Partition/Regionalization | Mesoscale Structure in | | |
|---|---|---|---|---|
| | | English WCN | German WCN | French WCN |
| English WCN | 28.42% | 100% | 35.29% | 33.26% |
| German WCN | 45.09% | 59.11% | 100% | 56.2% |
| French WCN | 46.33% | 60.9% | 53.14% | 100% |

$IFI_{xy}$ represents the extent to which a specific partition $x$ approximates the optimal partition $y$ detected by the weighted stochastic block model (WSBM), as defined in Equation (3). The *IFI* value ranges from 0 to 1.

These findings also highlight that different people may have different geographic imaginations of globalization. According to Table 2, the multicores–peripheries structure detected in the English WCN can only explain about 60% of the optimal mesoscale structures in the German and French WCNs. Using the optimal partition in the English world to understand the German and French worlds may bring about a 40% deviation. It might be biased if we believe that the networked structure perceived in the English world equals to that perceived in the world of other languages or cultures. This also demonstrates the importance of scrutinizing multiple globalizations and multiplex WCNs in different cultures and languages, rather than only focusing on the singular, uncontextualized network of economic linkages.

Additionally, the mesoscale structures in the German and French WCNs look more similar to each other than to the English WCN. The mesoscale structure in the German WCN is 35% similar to the English network and 53% to the French network, while that in the French WCN is 33% similar to the English WCN and 56% to the German network. These findings seemingly imply the asymmetrical influences of different languages in shaping WCNs. The mesoscale structure detected in the English network appears to have a better explanatory power to interpret the WCNs of other languages. This may relate to the fact that English is a more globalized language than German and French. Thus, the geographic imagination of globalization in the English world is more generalized; that is,

the generalization ability of the partition model in the English WCN is larger than that in German and French networks.

## 6. Conclusions

This study contributes to the debate on multiple globalizations by highlighting the multiscale structures of multiplex WCNs reflected in the Internet webpage content in different languages. We crawled the data with the Advanced Google Search Engine, which provides the searching service by language, and defined the weight of intercity linkages as the number of webpages with both cities' names in English, German, or French. These language-based networks differ from the WCNs in most previous studies with focuses on singular economic linkages, filling the gap of the WCN comparison by language and culture [9,10].

Relying on the big data sets, we adopted a complex-network approach by conducting a macroscale analysis of city rankings and distributions of local measures as well as a mesoscale analysis of detecting potential grouping structures, based on a weighted stochastic block model (WSBM). While most existing studies neglect comparison of the mesoscale structures in WCNs, our findings demonstrate the importance of discussing mesoscale properties in comparative WCN studies because the mesoscale analysis can deepen our understanding on the characteristics of WCNs that are not easily observed in the microscale or macroscale analysis [14]. Moreover, the results demonstrate the advantage of the WSBM approach, in contrast to traditional city clustering methods, such as principal component analysis, hierarchical clustering, and community detection methods [19,21–24,54–56]. WSBMs can avoid the trap of methodological determinism and extract multiple configurations of core–periphery or hybrid structures in different WCNs, the structure of which are difficult to be detected in other clustering methods. However, the WSBM approach uses a machine learning algorithm to approximate posterior distribution that requires continuous iteration of data, resulting in a heavy computational burden. The approach thus needs to be improved for analyzing large-scale networks with many city nodes.

In particular, from the macroscale analysis, the global city networks in different languages have some similarities as well as dissimilarities. For example, regardless of languages, the intercity connections are mainly distributed in between Europe, North America, and Asia, while the central influential cities with top degree centralities in the three WCNs are primarily Euramerican cities. New York always ranks top in the three WCNs, suggesting its dominant global influence crossing cultural and language boundaries. On the other hand, the specific rankings of city centralities vary significantly across languages, because they are affected by the geographic distribution of languages and the globalized levels of cities in different languages. It is worth noting that no Chinese or Japanese cities appear in the top 20 city lists of the three Western languages, although they play an increasingly important role in shaping the global economic network [4,57]. It is interesting to compare our findings with some existing city clustering studies based on economic, cultural, and political globalization perspectives [4,5]. For example, Taylor [5] found that in the economic globalization, in addition to New York, London, and Paris, Tokyo is often recognized as one of the top five. In cultural globalization, the top city is no longer New York, but London. In the political globalization, Geneva and Brussels often play dominate roles, while New York follows.

At the meso scale, results reveal variant grouping structures in the WCNs by language, including a multicores-peripheries structure in the English world, a duo-cores–peripheries structure in the German world, and a typical core–periphery structure in the French world. While New York and Paris are recognized as the global and macroregional centers, respectively, the roles and positions of many world cities in the WCNs vary across language. Based on the *IFI* estimates, we quantified and detected some interesting differences in the mesoscale structures by language. First, the English WCN looks more globalized, while the French and German worlds appear more territorial. Second, different languages may

exhibit different geographic imaginations of globalization. Using the mesoscale structure detected in the English WCN to understand the world in other languages may be biased. This finding is comparable to the WCN study based on the global Islamic finance corporates [31], which reported that the WCN under the Islamic culture differs from the geographic imagination in the Western countries. Third, the mesoscale structure detected in the English WCN appears having a larger generalization ability to partition other language's WCNs, suggesting that the English WCN might be a more generalized network. All these findings enrich our understandings of varying latent mesoscale structures in WCNs observed in different languages and cultures. Fourth, the city networks varying across languages may manifest the cultural proximity between cities with conjunct histories of immigrants, diasporas, and colonial ties [20]. For example, due to the past colonial history, although some cities are far away from each other, they still have the same languages, share a common cultural identity, and are thus closely connected in the virtual space and thus in the webpage-based city networks in this study.

Several improvements deserve further investigations. First, future work should compare the WCNs in more languages, particularly in Chinese and Japanese, given the cities in China and Japan have become critical command and control centers in the world. Second, more cities should be involved in future studies, particularly those important cities that are widely mentioned in native languages but not in the Western languages. Third, the intercity linkages data by the co-concurrent frequency on webpages might fail to tell the details of what these linkages are. The measure of intercity linkages can be improved by considering the typology of webpage contents, such as business news, political news, transport information, or others, as recently carried out by Salvini, Fabrikant, and Hu et al. [13,17]. The improvement enables us to further compare different types of WCNs on webpages by language [13], as well as to build a comparable framework between the document approach and the corporate and infrastructure approaches [14]. Fourth, it would be interesting to compare the noneconomic network by language with the traditional economic-dominant WCNs, such as those connected by corporate or infrastructure linkages. Finally, future work should explore the reasons for the formation of mesoscale structures. It is a difficult but important breakthrough point for network research.

**Author Contributions:** Conceptualization, Wenjia Zhang; Methodology, Wenjia Zhang and Jiancheng Zhu; Data processing and analysis, Wenjia Zhang and Jiancheng Zhu; Data curation, Wenjia Zhang; Writing—original draft preparation, Wenjia Zhang, Jiancheng Zhu. and Pu Zhao; Writing—review and editing, Wenjia Zhang, Jiancheng Zhu, and Pu Zhao; Visualization, Jiancheng Zhu; Supervision, Wenjia Zhang; Project administration, Wenjia Zhang; Funding acquisition, Wenjia Zhang. All authors have read and agreed to the published version of the manuscript.

**Funding:** This research was funded by Shenzhen Municipal Philosophy and Social Science Program, grant number: SZ2019B008, the Natural Science Foundation of Guangdong Province, grant number: 2020A1515010847, Peking University (Shenzhen) Future Urban Laboratory Techand Foundation, grant number: 201803, National Natural Science Foundation of China, grant number: 41801158, Shenzhen Municipal Basic Research Project (Free Exploration), grant number: JCYJ201803302153551891, and Shenzhen Municipal Natural Science Foundation, grant number: JCYJ20190808173611341.

**Institutional Review Board Statement:** Not applicable.

**Informed Consent Statement:** Not applicable.

**Data Availability Statement:** The data are available from https://github.com/zhangwenjia-pku/world-cities-data, accessed on 3 March 2021.

**Acknowledgments:** We thank Jean-Claude Thill for his valuable comments and for providing assistant in data collection, and Christopher Aicher for providing Matlab codes to run the WSBM.

**Conflicts of Interest:** The authors declare no conflict of interest.

## Appendix A

**Table A1.** City distribution across groups in the English WCN.

| Group | Cities | Roles |
|---|---|---|
| 1 (1) | New York | $\alpha$-Core |
| 2 (10) | London, Paris, Berlin, Rome, Chicago, San Francisco, Toronto, Melbourne, Sydney, New Delhi | $\beta$-Core |
| 3 (17) | Los Angeles, Boston, Miami, Dallas, Philadelphia, Houston, Washington DC, Atlanta, Charlotte, Seattle, Detroit, Minneapolis, Honolulu, Vancouver, Montreal, Birmingham, Amsterdam | Semi-periphery #1 (of $\alpha - \beta$-cores) |
| 4 (5) | Singapore, Beijing, Hong Kong, Shanghai, Tokyo | $\gamma$-Core |
| 5 (23) | Barcelona, Madrid, Milan, Lisbon, Dublin, Prague, Athens, Brussels, Edinburgh, Istanbul, Saint Petersburg, Budapest, Krakow, Sao Paulo, Buenos Aires, Santiago, Lima, Seoul, Bangkok, Chengdu, Doha, Auckland, Cape Town | Semi-periphery #2 (of $\gamma$-Core) |
| 6 (17) | Munich, Frankfurt, Hamburg, Stuttgart, Cologne, Bonn, Moscow, Mumbai, Chennai, Bangalore, Dubai, Kolkata, Hyderabad, Ahmedabad, Osaka, Kobe, Cairo | Periphery#1 (of all cores) |
| 7 (18) | Lyon, Stockholm, Zurich, Geneva, Rotterdam, Vienna, Copenhagen, Oslo, Basel, Tel Aviv, Warsaw, Rio de Janeiro, Mexico City, Monterrey, Panama City, Johannesburg, Durban, Manila | Periphery#2 (of all cores) |
| 8 (35) | Taipei, Guangzhou, Tianjin, Shenzhen, Chongqing, Dalian, Kuala Lumpur, Jakarta, Abu Dhabi, Nagoya, Hanoi, Karachi, Beirut, Tehran, Colombo, Busan, Almaty, Ho Chi Minh City, Riyadh, Bandung, Dhaka, Jeddah, Guadalajara, Belo Horizonte, Porto Alegre, Bogota, Caracas, Medellin, Bucharest, Kiev, Dusseldorf, Ankara, Alexandria, Nairobi, Lagos | Periphery#3 (of all cores) |

Note: Colors represent groups in Figure 3.

**Table A2.** City distribution across groups in the German WCN.

| Group | Cities | Roles |
|---|---|---|
| 1 (1) | New York | $\alpha$-Core |
| 2 (16) | Berlin, Bonn, Budapest, Cologne, Delhi, Dusseldorf, Frankfurt, Hamburg, London, Mumbai, Munich, Paris, Rome, Sao Paulo, Stuttgart, Zurich | $\beta - $Core |
| 3 (31) | Abu Dhabi, Amsterdam, Atlanta, Bangkok, Barcelona, Beijing, Birmingham, Brussels, Charlotte, Chongqing, Detroit, Doha, Dubai, Dublin, Edinburgh, Geneva, Honolulu, Houston, Istanbul, Madrid, Miami, Minneapolis, Philadelphia, Rotterdam, San Francisco, Seattle, Shanghai, Singapore, Tianjin, Toronto, Washington DC | Periphery #1 (of $\alpha - $Core) |
| 4 (44) | Alexandria, Ankara, Athens, Auckland, Bucharest, Buenos Aires, Cairo, Cape Town, Caracas, Chengdu, Copenhagen, Durban, Guadalajara, Guangzhou, Jakarta, Johannesburg, Kiev, Krakow, Kuala Lumpur, Lima, Lisbon, Los Angeles, Lyon, Manila, Melbourne, Mexico City, Montreal, Moscow, Nairobi, Osaka, Oslo, Rio de Janeiro, Riyadh, Saint Petersburg, Seoul, Shenzhen, Stockholm, Sydney, Taipei, Tehran, Tel Aviv, Tokyo, Vancouver, Warsaw | Periphery #2 (of $\alpha - $Core) |
| 5 (7) | Basel, Boston, Chicago, Dallas, Hong Kong, Milan, Vienna | Periphery #3 (of $\beta - $Core) |
| 6 (27) | Ahmedabad, Almaty, Bandung, Bangalore, Beirut, Belo Horizonte, Bogota, Busan, Chennai, Colombo, Dalian, Dhaka, Hanoi, Ho Chi Minh City, Hyderabad, Jeddah, Karachi, Kobe, Kolkata, Lagos, Medellin, Monterrey, Nagoya, Panama City, Porto Alegre, Prague, Santiago | Periphery #4 (of $\alpha - $Core) |

Note: Colors represent groups in Figure 3.

**Table A3.** City distribution across groups in the French WCN.

| Group | Cities | Roles |
|---|---|---|
| 1 (1) | New York | α-Core |
| 2 (2) | Hong Kong, Paris | β-Core |
| 3 (14) | Amsterdam, Berlin, Budapest, Cologne, Copenhagen, London, Lyon, Miami, Milan, Montreal, Rome, Seattle, Stockholm, Vancouver | γ-Core |
| 4 (27) | Athens, Birmingham, Boston, Brussels, Charlotte, Chicago, Dallas, Delhi, Dublin, Frankfurt, Hamburg, Houston, Istanbul, Los Angeles, Melbourne, Moscow, Munich, Seoul, Shanghai, Singapore, Stuttgart, Sydney, Tokyo, Toronto, Vienna, Warsaw, Zurich | Periphery#1 (of all cores) |
| 5 (39) | Ahmedabad, Ankara, Atlanta, Auckland, Bangalore, Bangkok, Basel, Beijing, Bonn, Buenos Aires, Chengdu, Chennai, Detroit, Dusseldorf, Geneva, Guadalajara, Guangzhou, Honolulu, Johannesburg, Kiev, Krakow, Kuala Lumpur, Madrid, Mexico City, Minneapolis, Mumbai, Oslo, Panama City, Philadelphia, Prague, Rio de Janeiro, Rotterdam, Saint Petersburg, San Francisco, Santiago, Sao Paulo, Shenzhen, Taipei, Washington DC | Periphery#2 (of all cores) |
| 6 (43) | Abu Dhabi, Alexandria, Almaty, Bandung, Barcelona, Beirut, Belo Horizonte, Bogota, Bucharest, Busan, Cairo, Cape Town, Caracas, Chongqing, Colombo, Dalian, Dhaka, Doha, Dubai, Durban, Edinburgh, Hanoi, Ho Chi Minh City, Hyderabad, Jakarta, Jeddah, Karachi, Kobe, Kolkata, Lagos, Lima, Lisbon, Manila, Medellin, Monterrey, Nagoya, Nairobi, Osaka, Porto Alegre, Riyadh, Tehran, Tel Aviv, Tianjin | Periphery#3 (of all cores) |

Note: Colors represent groups in Figure 3.

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
