# Peer review of "Comparing World City Networks by Language: A Complex-Network Approach"

_ijgi, doi:10.3390/ijgi10040219_

Round 1

Reviewer 1 Report

Excellent work and fairly innovative overall approach !!!

Very good presentation of methods and results also.

Minor linguistic suggestions such as 

line 149 "We took advantage"

line 400 "findings also illuminate" (indicative suggestion: change to "highlight")

could be improved maybe by an external editor.

As far as the assumption "that if two cities appear on the same webpage, it implies that they have a connection", may be it could be rephrased.

e.g. The appearance of two or more cities in a web page may not always imply a connection, but it is sure a kind of conceptual link under a general  cognitive linguistic framework. (I am not sure for the exact selection of terms or syntax, but  i think it needs a more elaborated "phrasing")

Conclusion and Discussion section, set the overall advantages and disadvantages for the methodological approach and hypothesis.

Reviewer 2 Report

  1. The authors present a complex-network approach for analyzing the multiscale structure of world city network based on the collected web page data according to three different languages, namely, English, French and German. Its major contribution focuses on the use of quantitative measures at the mesoscale of WCN, such that the pattern of WCN can be analyzed and the similarity between different WCNs can be compared. It also provides the advantage of avoiding biases introduced by the various models from the past research.
  2. The discussion about past research is comprehensive, widely covering different viewpoints and perspectives for the research of WCN.
  3. The major results are based on the cooccurrence of two cities in the collected web pages. While the collected volume of webpages is surely impressive, the value or meanings for the “cooccurrence of cities” data need to be further justified, as no particular perspective or constraint is taken into consideration. To be more specific, what can we infer if two cities are mentioned in the same web page?
  4. Can the city Nodes (Sizes of markers) in Figure 1 be further discussed on the basis of quantitative values? (e.g., the values in Table 1 for the city of New York in different languages appear to be rather different, but the size of map symbols in Figure 1 are similar).
  5. The WCN analysis at the mesoscale in the research may be related to the first languages mainly used for the cities in their respective countries. In the results (e.g., appendix A), is the issue where the cities in the same group use the same first language considered in this research?
  6. How to determine the number of cores and the core-periphery structure in the WCN?
  7. The discussion of continent-based regionalization in line 394-396 is rather interesting, how is the value of “50%” determined?
  8. The WCN analysis based on language in this paper is rather interesting. Although the authors mention to consider the addition of “typology” of the webpage content in the future research, it would be interesting to compare the results with another globalization WCN analysis based on other perspectives (e.g. culture, business, social/economic) to demonstrate the distinguished contribution of the research outcome (the comparison can be from other research). 

Reviewer 3 Report

In this article, the authors compare World City Networks by Language through a Complex Network Approach. The paper is well written and easy to read.

In order to improve the paper, I recommend the following suggestions to the authors:

The introduction is too long. I suggest the authors split it into two sections: Introduction and Related Work.

Section 2. The authors could provide an example of extracted data.

Section 3. It is not clear which are the inputs and the outputs of the employed models.

Section 3, line 188 the authors should provide a definition of degree centralities, since this measurement is exploited in the next sections.

Section 4. line 322 the authors should provide a definition of weight distributions.

Fig. 4 is not clear. For example the authors could provide a label for each axis. I guess that each graph represents the English, French and German situation. The authors could add this aspect.

Author Response

Response to reviewer 1
